# Structure-guided engineering enables E3 ligase-free and versatile protein ubiquitination via UBE2E1

Xiangwei Wu [1,2,5], Yunxiang Du[1,5], Lu-Jun Liang [3,5] ✉, Ruichao Ding[1], Tianyi Zhang [1], Hongyi Cai[1], Xiaolin Tian[4], Man Pan [2] ✉ & Lei Liu [1] ✉

Ubiquitination, catalyzed usually by a three-enzyme cascade (E1, E2, E3), regulates various eukaryotic cellular processes. E3 ligases are the most critical components of this catalytic cascade, determining both substrate specificity and polyubiquitination linkage specificity. Here, we reveal the mechanism of a naturally occurring E3-independent ubiquitination reaction of a unique human E2 enzyme UBE2E1 by solving the structure of UBE2E1 in complex with substrate SETDB1-derived peptide. Guided by this peptide sequence-dependent ubiquitination mechanism, we developed an E3-free enzymatic strategy SUE1 (sequence-dependent ubiquitination using UBE2E1) to efficiently generate ubiquitinated proteins with customized ubiquitinated sites, ubiquitin chain linkages and lengths. Notably, this strategy can also be used to generate site-specific branched ubiquitin chains or even NEDD8-modified proteins. Our work not only deepens the understanding of how an E3-free substrate ubiquitination reaction occurs in human cells, but also provides a practical approach for obtaining ubiquitinated proteins to dissect the biochemical functions of ubiquitination.

The covalent attachment of ubiquitin (Ub) to substrates, referred to as ubiquitination, represents one of the most versatile and prevalent posttranslational modifications (PTMs) regulating almost every process in eukaryotic cells, including protein degradation, DNA repair, and receptor transport[1]. Deficiencies in the ubiquitin system are implicated in many human pathologies, such as cancer, immune defects, and neurodegeneration[2,3]. Typically, a three-enzyme cascade consisting of the E1 Ub-activating enzyme, the E2 Ub-conjugating enzyme, and the E3 Ub-protein ligase, catalyzes the coupling of the C-terminal carboxylate of Ub to a lysine within the substrate protein[4].

Mechanistic studies of various E3 ligases by colleagues and us revealed that E3 ligases are the critical components of these Ub cascades (E1/E2/E3s) owing to their selection of substrate proteins, modification sites on those proteins, and the types of Ub chains[5–11]. Therefore, E3 ligases are key to accessing ubiquitinated proteins to understand the effects of ubiquitination on specific substrates. However, thus far, only a few substrate proteins have defined Ub E3 ligases, and consequently, only a small subset of ubiquitination events has been studied in detail, which has been a major obstacle in understanding cellular events regulated by ubiquitination[12,13].

Given that identifying the E3s for target proteins is a key challenge in reconstituting ubiquitinated substrates, as described above, elucidating the underlying mechanism of this extraordinary E3-free Ub cascade occurring in human cells not only will improve our

[1]New Cornerstone Science Laboratory, Tsinghua-Peking Joint Center for Life Sciences, MOE Key Laboratory of Bioorganic Phosphorus Chemistry and Chemical Biology, Center for Synthetic and Systems Biology, Department of Chemistry, Tsinghua University, Beijing 100084, China. [2]Institute of Translational Medicine, School of Chemistry and Chemical Engineering, School of Pharmacy, National Center for Translational Medicine (Shanghai), Shanghai Jiao Tong University, Shanghai 200240, China. [3]Center for BioAnalytical Chemistry, Hefei National Laboratory of Physical Science at Microscale, University of Science and Technology of China, Hefei 230026, China. [4]MOE Key Laboratory of Bioinformatics, School of Life Sciences, Tsinghua University, 100084 Beijing, China. [5]These authors contributed equally: Xiangwei Wu, Yunxiang Du, Lu-Jun Liang. ✉e-mail: lujun@ustc.edu.cn; panman@sjtu.edu.cn; lliu@mail.tsinghua.edu.cn

understanding of the ubiquitination reaction evolved from nature, but also may provide useful strategies for the acquisition of ubiquitinated proteins[14]. To this end, we focused on a unique human ubiquitin-conjugating enzyme E2 E1 (UBE2E1) that can catalyze mono-ubiquitination at K867 of its substrate protein SETDB1 independently of E3 through an unknown mechanism[15].

Herein, we report the crystal structure of UBE2E1 in complex with the substrate SETDB1-derived peptide. Our biochemical and structural studies demonstrate the mechanism by which UBE2E1 specifically recognizes and ubiquitinates the peptide KEGYES. Through structure-guided optimization, we obtained a peptide KEGYEE with higher ubiquitination efficiency by UBE2E1, and by introducing this peptide sequence into a given substrate, we efficiently conjugated monoUb, diUb with different linkage, polyUb, polyUb mixture, branched Ub chains and NEDD8 to defined lysine residues on the substrate in an E3-free manner.

## Results

### Structural insights into SETDB1-derived peptide ubiquitination by UBE2E1

To examine the mechanism underlying UBE2E1-mediated ubiquitination of the evolutionarily conserved hexapeptide in SETDB1 ([867]KEGYES[872])[15], we first reconstituted the ubiquitination reaction using purified proteins, including Uba1, UBE2E1, and a model substrate EGFP* bearing the hexapeptide (KEGYES) at its C-terminus. We observed that EGFP* was monoubiquitinated, whereas almost no ubiquitination was observed for EGFP** bearing an inactive hexapeptide (REGYES) (Fig. 1a). MS/MS analysis confirmed that ubiquitination occurred on the lysine of the hexapeptide (Fig. 1b). The removal of ATP and $Mg^{2+}$, which are required for Uba1 to activate Ub or the mutation of the UBE2E1 catalytic cysteine to alanine both completely abolished the ubiquitination of EGFP (Fig. 1a).

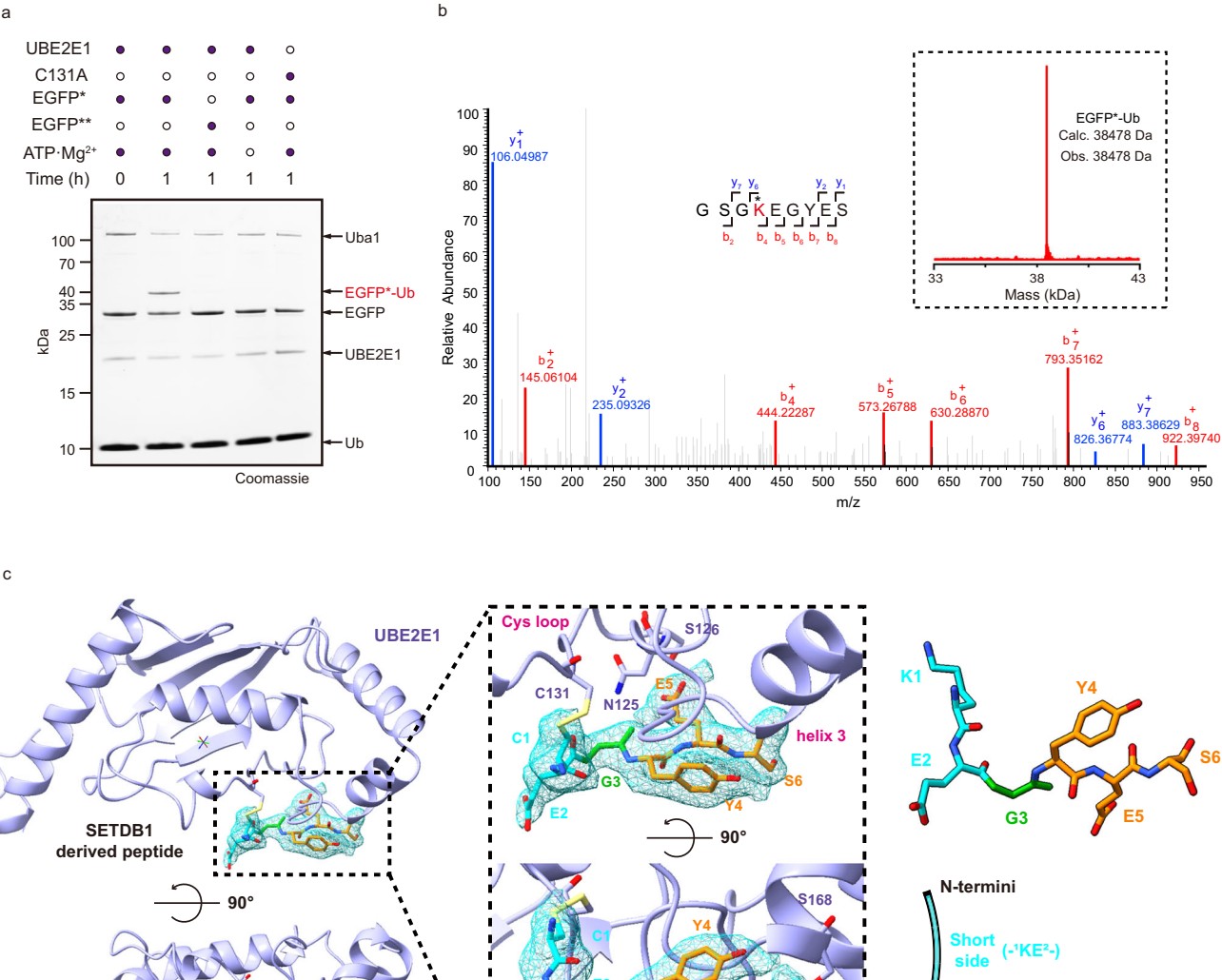

**Fig. 1 | Structure of the E2 enzyme UBE2E1 in complex with a SETDB1-derived peptide. a** In vitro UBE2E1-dependent ubiquitination assay using hexapeptide (KEGYES)-fused EGFP as substrates. Gel images are representative of independent biological replicates (n = 2). **b** Mass spectrometry confirms the ubiquitination product (EGFP*-Ub) and the lysine on the hexapeptide is the ubiquitination site. The observed (obs.) and calculated (calc.) molecular weights were marked. **c** Overall structure of UBE2E1 bound to hexapeptide. UBE2E1 is shown as ribbon and colored in bright blue. The peptide is shown as sticks: the long side, corner, and the short side of the "L shape"-hexapeptide are colored in bright orange, green and cyan, respectively. Source data are provided as a Source Data file.

To elucidate the molecular details governing hexapeptide recognition by UBE2E1, we initially attempted to cocrystallize UBE2E1 with the hexapeptide but only obtained apo UBE2E1 crystals. We then cross-linked the catalytic cysteine of UBE2E1 to the hexapeptide via a disulfide bond, in which the lysine in the hexapeptide was mutated to cysteine to enable the reaction[14,16] (Supplementary Fig. 1a–c, Supplementary Fig. 11). The crystal structure of the hexapeptide crosslinked with UBE2E1 was obtained and found to diffract to 2.43 Å resolution (Fig. 1c, Supplementary Fig. 1d, e, Supplementary Table 1). When the structure of UBE2E1 in our complex was superimposed with the structure of apo UBE2E1 reported previously[17] (PDB: 3BZH), no significant conformational changes were observed (the root-mean-square deviation (RMSD) is 0.405 Å over 140 Cα). The hexapeptide is located between the loop containing the active cysteine (Cys loop) and a loop preceding helix 3(Fig. 1c)[18]. An L-shaped structure is employed for this peptide, where the 1st lysine (K1) and 2nd glutamate (E2) together constitute the short side of the "L", followed by the 3rd glycine (G3) at the corner, and the remaining three residues (-$^4$YES$^6$-) on the long side of the "L" (Fig. 1c).

The long side of the "L"-hexapeptide is responsible for the major interaction with UBE2E1, with an interaction area of 445 Å$^2$, and extensive interactions involving the main chains were observed in this interface. Specifically, the 4th tyrosine (Y4) and 5th glutamate (E5) act as two anchor points by binding P164 and S126 of UBE2E1 through hydrophobic and hydrogen bonding interactions, respectively. (Fig. 2a). The amino and the backbone carbonyl groups of E5 form hydrogen bonds with the carbonyl group of UBE2E1 L165 and the amino group of G167, respectively (Fig. 2a). The carbonyl group of S6 engages in hydrogen bonds with S168 of UBE2E1, further stabilizing the interaction (Fig. 2a). To validate the importance of the above interactions, a series of mutated hexapeptide-tagged EGFP or mutated UBE2E1 proteins were generated and subjected to E3-free ubiquitination assays to examine substrate ubiquitination efficiency. The Y4A and E5A mutations in hexapeptide both dramatically decreased ubiquitination efficiency (Supplementary Fig. 2a). Correspondingly, either P164A or S126A mutations in UBE2E1 abrogated substrate ubiquitination (Supplementary Fig. 2b). These results together highlight the importance of anchor points in mediating hexapeptide recognition. To further verify the contribution of the backbone to recognition, a hexapeptide-tagged ubiquitin variant with the backbone nitrogen atom of E5 methylated was chemically synthesized and used as the substrate (Supplementary Fig. 12). As expected, disruption of this interaction caused a remarkable decrease in ubiquitination efficiency (Supplementary Fig. 2c, d).

At the corner of the "L"-hexapeptide, the carbonyl group of G3 forms a hydrogen bond with UBE2E1 N125, and as expected, the N125A mutant of UBE2E1 failed to catalyze substrate ubiquitination (Fig. 2a, Supplementary Fig. 2b). Notably, the G3A mutation greatly reduced the efficiency of ubiquitination (Supplementary Fig. 2a), suggesting that glycine is critical at position 3. Regarding the short side of the "L"-hexapeptide, K1 (Cys1 in the crystal structure) is proximal to D163 of UBE2E1 (Fig. 2b), which is conserved among other E2 enzymes and proposed to activate the nucleophilicity of substrate lysine residues for ubiquitin conjugation[19,20], whereas no obvious interaction was observed between position 2 glutamate and UBE2E1. Interestingly, introducing Lys on EGFP* at position 0 or 2 does not lead to ubiquitination of mutants, suggesting that an appropriate distance between K1 and G3 is necessary to attack the E2-Ub thioester (Fig. 2d, Supplementary Fig. 2e). Furthermore, Hydrogen-deuterium exchange mass spectrometry (HDX-MS) was employed to validate the interaction between the UBE2E1 and the hexapeptide in solution (Supplementary Table 5). In the presence of the hexapeptide, a significant decrease in the deuterium exchange level was observed for the UBE2E1 (116–132) fragment (Supplementary Fig. 3), which is close to the catalytic

cysteine and contains the key amino acids N125/S126 for the interaction observed in the crystal structure.

In addition, through sequence alignment, we found that the residues of UBE2E1 involved in hexapeptide recognition are not conserved in other ubiquitin E2s (Supplementary Fig. 2f), consistent with the inability of other E2 enzymes to mediate ubiquitination of SETDB1 containing native hexapeptide sequences[15]. For example, some residues (N125/S126/P164/L165) are conserved in Ubch5c, while another two important residues (G167/S168) are replaced by P121/E122. To our surprise and delight, the introduction of P121G/E122S mutations into Ubch5c enabled it to mediate substrate site-specific ubiquitination in an E3-free manner, whereas wild-type Ubch5c did not (Fig. 2e).

Collectively, our structural and biochemical results revealed that UBE2E1 specifically recognizes the hexapeptide in an "L" shape and localizes it near its active site for ubiquitin transfer in an E3-free mechanism. (Fig. 2c).

## Structure-guided optimization of sequence-dependent ubiquitination by UBE2E1 for the generation of ubiquitinated substrates

Guided by the recognition mechanism from our complex structure, where E2/S6 side chains are not involved in the interaction with UBE2E1, we next experimentally mutated these two positions to optimize a sequence with higher ubiquitination efficiency. Seven amino acids(L/F/S/E/R/Q/G) with different side chain properties were introduced at position 2 or 6 of the hexapeptide-tagged EGFP, and these mutants were then applied to the E3-free ubiquitination assay. All mutants were ubiquitinated as expected (Supplementary Fig. 4a, b), and encouragingly, the S6E mutation increased the ubiquitination efficiency from 60% to 90% in 2 h (Supplementary Fig. 4b). The $K_m$, $k_{cat}$ values of UBE2E1 towards the original hexapeptide and the S6E peptide were further measured using a standard protocol based on Lineweaver-Burk plots (Supplementary Table. 2), and the kinetic data showed that the S6E mutation increased the ubiquitination activity from 19.60 to 38.29 M$^{-1}$ s$^{-1}$ and shortened the half-life of the reaction by 2.9-fold (from 1.36 h to 0.47 h) under our conditions (Supplementary Fig. 4c).

Based on this optimized sequence (-KEGYEE-), we developed an E3-free enzymatic strategy for the generation of ubiquitinated proteins with customized ubiquitinated sites, ubiquitin chain linkages and lengths, termed sequence-dependent ubiquitination using UBE2E1 (SUE1) (Fig. 3a). To demonstrate the versatility of the SUE1 strategy in the generation of ubiquitinated substrates, we introduced a SUE1 tag (-KEGYEE-) into different parts of the substrate EGFP, including the N-terminus (EGFP-N), internal region (EGFP-I) and C-terminus (EGFP-C), and the expressed EGFP variants were subjected to an in vitro monoubiquitination assay using UBE2E1 (Fig. 3b). Ubiquitination formed on all EGFP variants with near-quantitative conversion (>90% as determined by gel density) and this ubiquitination was all dependent on the lysine residues within the SUE1 tag (Fig. 3b).

We next examined the ability of the SUE1 strategy to transfer polyUb chains to the substrate[21] (Fig. 3c). Eight types of diUb (M1, K6, K11, K27, K29, K33, K48, K63-linked) were applied as donor Ub-chain sources of the ubiquitination reaction, and we observed that all diUb was quantitatively transferred onto the SUE1 tag in EGFP-C (Fig. 3c, Supplementary Fig. 5a). Longer Ub chains, including K29-linked or K48-linked triUb, tetraUb, and pentaUb, were also used to modify EGFP-C through our strategy, and the reactions achieved almost quantitative conversion (Fig. 3d, Supplementary Fig. 5b). Taking the preparation of K48 tetraUb-modified EGFP-C as an example, the product reached the milligram level, and LC–MS confirmed its correctness (Supplementary Fig. 5c, d). Moreover, the mixture of K48-linked polyUb chains prepared by UBE2K was

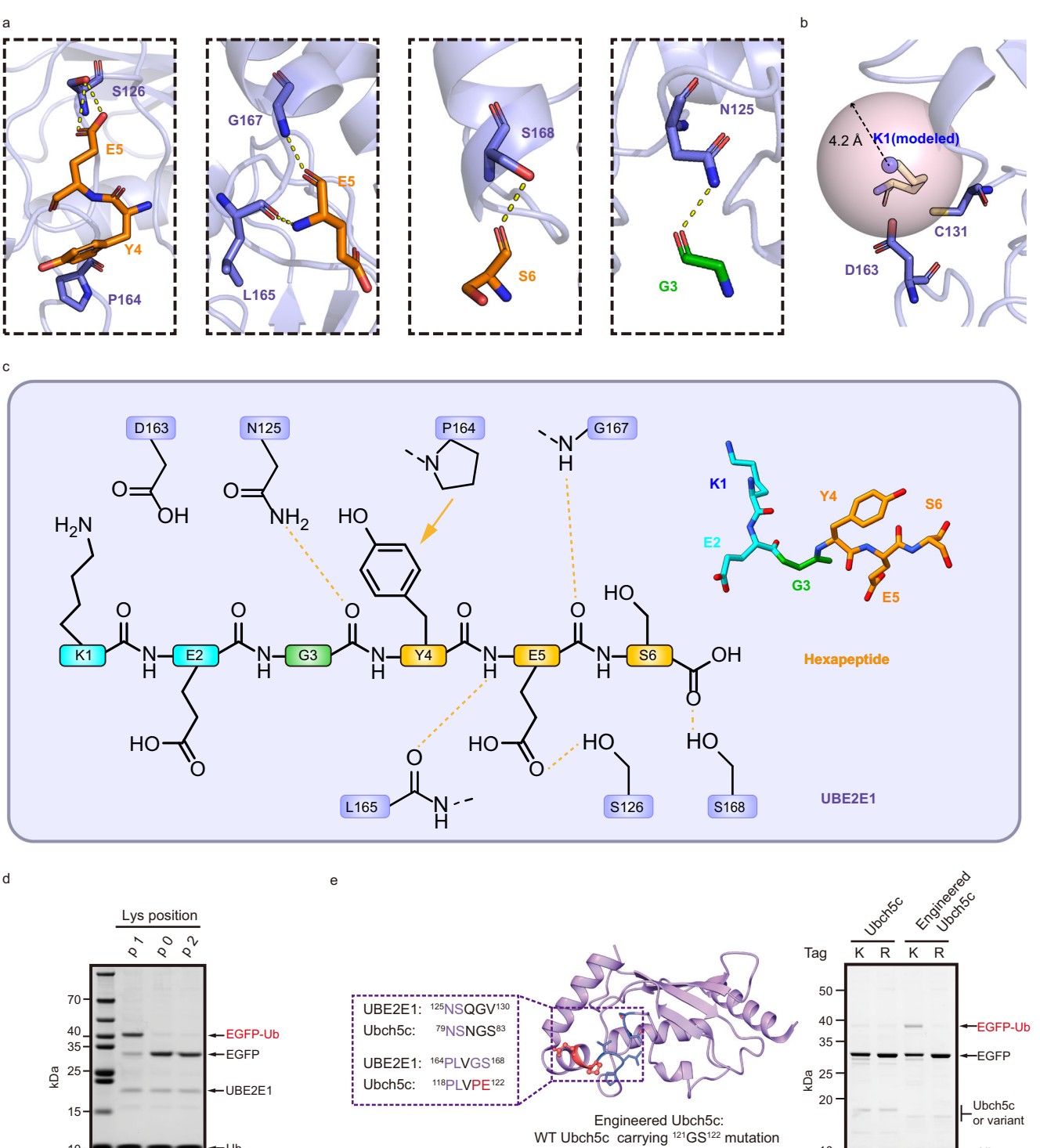

**Fig. 2 | Molecular insights into UBE2E1-mediated ubiquitination without E3.**
**a** Detailed interactions of hexapeptide with UBE2E1 residues. The intermolecular hydrogen bonds are indicated with yellow dashed and the residues are shown as sticks with the same color code as in Fig. 1c. **b** UBE2E1 D136 residue poised to activate the modeled lysine residue's nucleophilicity. **c** Interaction diagram between UBE2E1 and SEDTB1-derived peptide. Intermolecular hydrogen bonds and hydrophobic interactions are indicated with yellow dashes and arrows, respectively. **d** The proper distance between the receptor lysine and glycine at the corner of "L-shaped" hexapeptide is necessary for ubiquitination. **e** Engineered Ubch5c enabled site-specific ubiquitination in an E3-free manner. Gel images shown in (**d**) and (**e**) are representative of independent biological replicates (*n* = 2). Source data are provided as a Source Data file.

directly applied in the SUE1 strategy, and again, successful site-specific transferring of polyUb chains onto EGFP-C in situ was observed (Supplementary Fig. 6a, b). Additionally, we further tested the SUE1 strategy in generating K11/48-branched Ub chain-modified protein. Specifically, we first prepared K11/48-branched triUb by

protein chemical synthesis[22,23] (Supplementary Fig. 6c) and then performed an E3-free ubiquitination reaction to ligate K11/48-branched triUb and substrate EGFP-C. To our delight, after 12 h, the K11/48-branched triUb was transferred to the substrate in 78% yield (Fig. 3e).

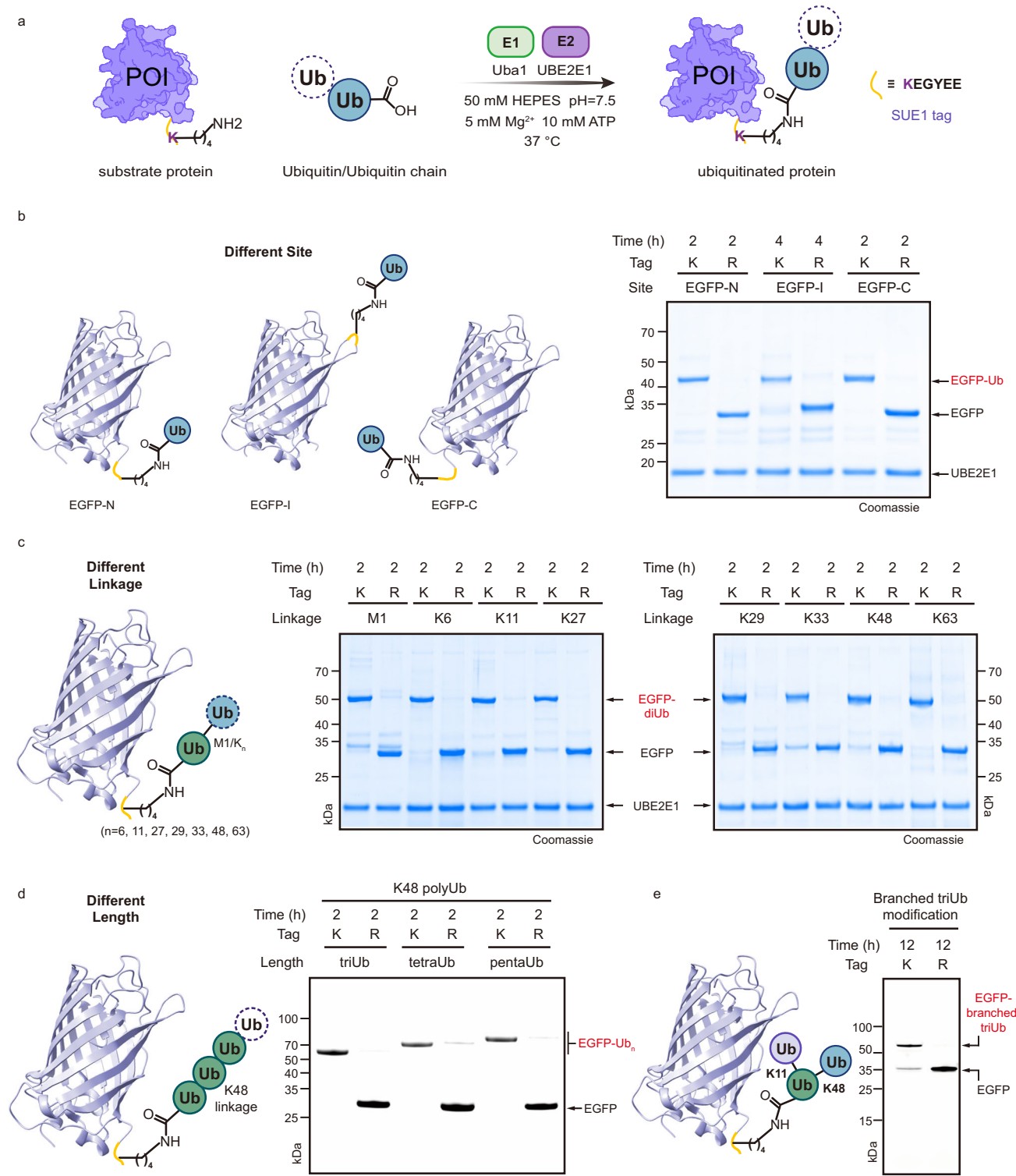

**Fig. 3 | Exploring sequence-dependent ubiquitination by UBE2E1 (SUE1) as a general strategy for protein ubiquitination. a** Schematic representation of Uba1/UBE2E1-mediated site-specific ubiquitination, in which the protein of interest (POI) bears the SUE1 tag (KEGYEE) at the customized site and monoUb or ubiquitin chain is used as the ubiquitin source. Generally, 1 μM Uba1, 20 μM UBE2E1, 8 μM substrate containing the SUE1 tag, 80 μM monoUb or 40 μM Ub chain were mixed, reacted in the reaction buffer (50 mM HEPES, pH 7.5, 150 mM NaCl, 5 mM MgCl₂ and 10 mM ATP) at 37 °C for 2 h. **b** The SUE1 strategy for ubiquitination at different sites, the N-terminus, internal region or C-terminus of EGFP was chosen to introduce a SUE1 tag, respectively. **c** The SUE1 strategy for ubiquitination with different linkages of Ub chains, M1/K6/K11/K27/K29/K33/K48/K63-linked diUb was used to transfer to the substrate respectively. **d** The SUE1 strategy for ubiquitination with different lengths of Ub chains, taking K48-linked Ub chains as an example. **e** The SUE1 strategy for ubiquitination with branched Ub chain (K11/K48 branched). Gel images shown in (**b–e**) are representative of independent biological replicates (*n* = 2). Source data are provided as a Source Data file.

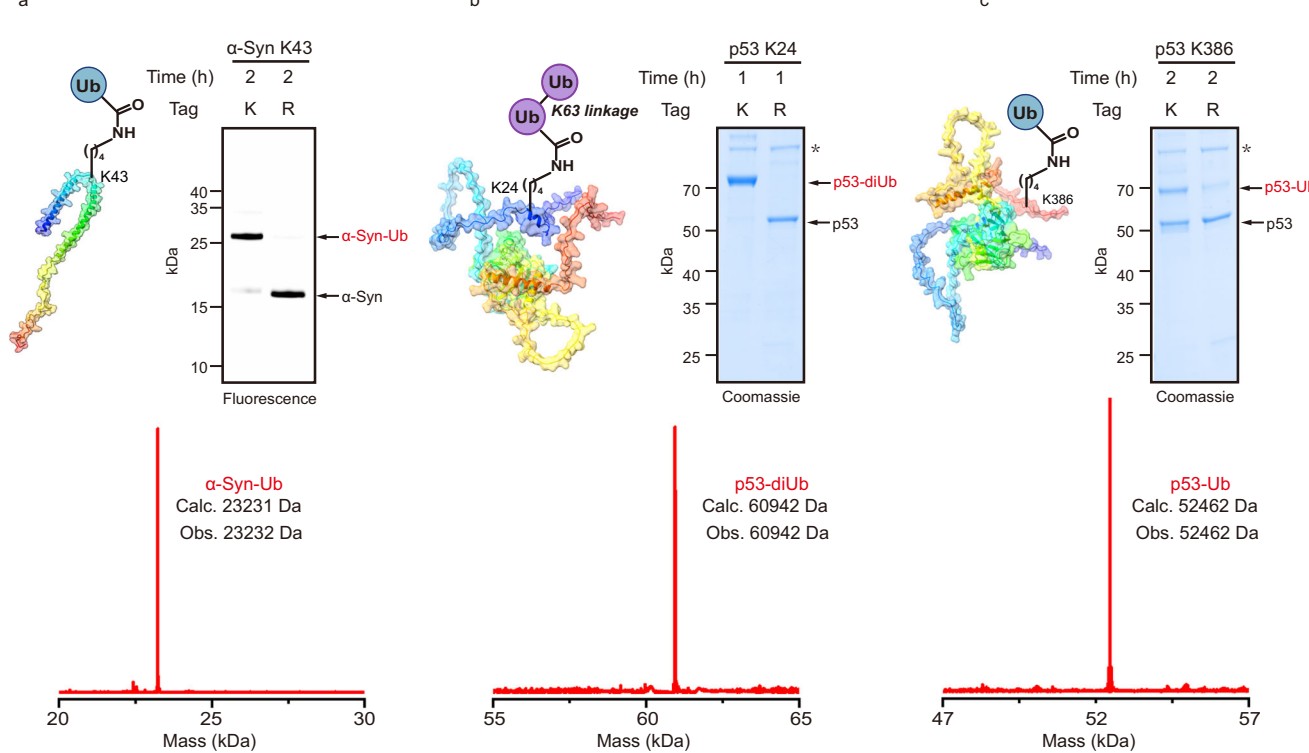

**Fig. 4 | Ubiquitination of authentic substrate protein via the SUE1 strategy.**
**a** Top, SDS–PAGE analysis of SUE1-mediated monoubiquitination of α-Synuclein (α-Syn) at K43 using In-gel fluorescence for clear visualization of ubiquitination. The model of α-Synuclein was derived from the NMR structure of the wild-type α-Synuclein (PDB: 1XQ8). Bottom, deconvoluted ESI-MS of the purified mono-ubiquitinated α-Synuclein (α-Syn-Ub). **b** Top, SDS–PAGE analysis of SUE1-Mediated K63-linked ubiquitin chain modification of p53 at K24 and asterisk (*) denotes an impurity in the p53-stock. The p53 model was derived from the predicted structure of wild-type p53 in the AlphaFold Protein Structure Database (AF-P04637-F1-model_v4). Bottom, deconvoluted ESI-MS of the purified diubiquitinated p53 (p53-diUb). **c** Top, SDS–PAGE analysis of SUE1-mediated monoubiquitination of p53 at K386 and asterisk (*) denotes an impurity in the p53-stock. Deconvoluted ESI-MS of the purified ubiquitinated products. Bottom, deconvoluted ESI-MS of the purified monoubiquitinated p53 (p53-Ub). Gel images shown in (**a**–**c**) are representative of independent biological replicates (*n* = 2). Source data are provided as a Source Data file.

Encouraged by the successful application of the SUE1 strategy in preparing ubiquitinated EGFP variants, we next approached the challenge of obtaining biologically relevant ubiquitinated proteins with the SUE1 strategy. We first aimed to generate mono-ubiquitinated α-synuclein at Lys43, which was reported to inhibit the formation of α-synuclein fibrils[24]. Mutations were introduced to provide a SUE1 tag at Lys43 of α-synuclein, followed by a reaction catalyzed by UBE2E1, and specific monoubiquitination at Lys43 was observed with >90% conversion measured by in-gel fluorescence densitometry (Fig. 4a, Supplementary Fig. 7a). Ubiquitinated tumor suppressor protein p53 was chosen as the second example, in which K63-linked ubiquitination on p53 at Lys24 is reported to restrict mitochondrial localization and spontaneous apoptosis[25]. With the same procedures mentioned above, K63-linked diUb-modified p53 was obtained with an efficiency of >90%, and MS/MS confirmed that the modification site was Lys24 (Fig. 4b, Supplementary Fig. 7b). Likewise, monoubiquitinated p53 at K386, which contributes to p53 nuclear export[26], was achieved with 43% conversion by introducing a SUE1 tag (Fig. 4c, Supplementary Fig. 7c). For the third example, we prepared the K48-linked diUb-modified N-terminus of cyclin B1(NCB1) at K64[27,28], which is a classical substrate for biochemical and structural studies of the proteasome, and the reaction was successfully completed with a conversion rate of more than 70% (Supplementary Fig. 6d). Taken together, these results demonstrate the flexibility of the SUE1 strategy in generating various ubiquitinated substrates with defined sites, ubiquitin chain linkages and lengths.

## SUE1-assisted synthesis of multisite ubiquitinated protein

Ubiquitination often occurs simultaneously at several lysine residues for many substrate proteins. For example, α-synuclein was reported to be ubiquitinated at multiple sites with monoUb, K48-linked or K63-linked Ub chains, regulating different cell fates of α-synuclein[29]. Advances in total or semi-synthesis of protein chemistry have recently facilitated access to single-site ubiquitinated α-synuclein and have elucidated how ubiquitinated sites and ubiquitin chain length regulate α-synuclein aggregation[24,30,31]. However, the synthesis of multisite ubiquitinated α-synuclein has remained challenging, and the effect of this ubiquitinated form on α-synuclein aggregation remains unclear. To this end, we examined whether the SUE1 strategy could be used to access multisite ubiquitinated proteins. Consequently, two SUE1 tags were introduced to α-synuclein at Lys43 and Lys96, and >90% conversion to the doubly monoUb-modified product was observed within 2 h under similar conditions to the single-site ubiquitination reaction (Fig. 5a, b).

Furthermore, to generate multisite ubiquitinated α-synuclein with different Ub chain linkages, we applied our SUE1 ubiquitination strategy in combination with the LACE strategy, an enzymatic strategy based on engineering Ub-E1 and SUMO-E2 to enable ubiquitin attachment to LACE-tagged proteins[14,21] (Fig. 5c). First, we investigated whether UBE2E1 and Ubc9 K14R-mediated substrate ubiquitination interfered with each other. We found that UBE2E1 catalyzes the ubiquitination of SUE1-tagged EGFP but not LACE-tagged EGFP, and Ubc9 K14R does the opposite (Supplementary Fig. 8a), which means that UBE2E1 and Ubc9 K14R constitute a bidirectionally orthogonal enzyme pair that can be used to ligate Ub or the Ub chain to different sites of

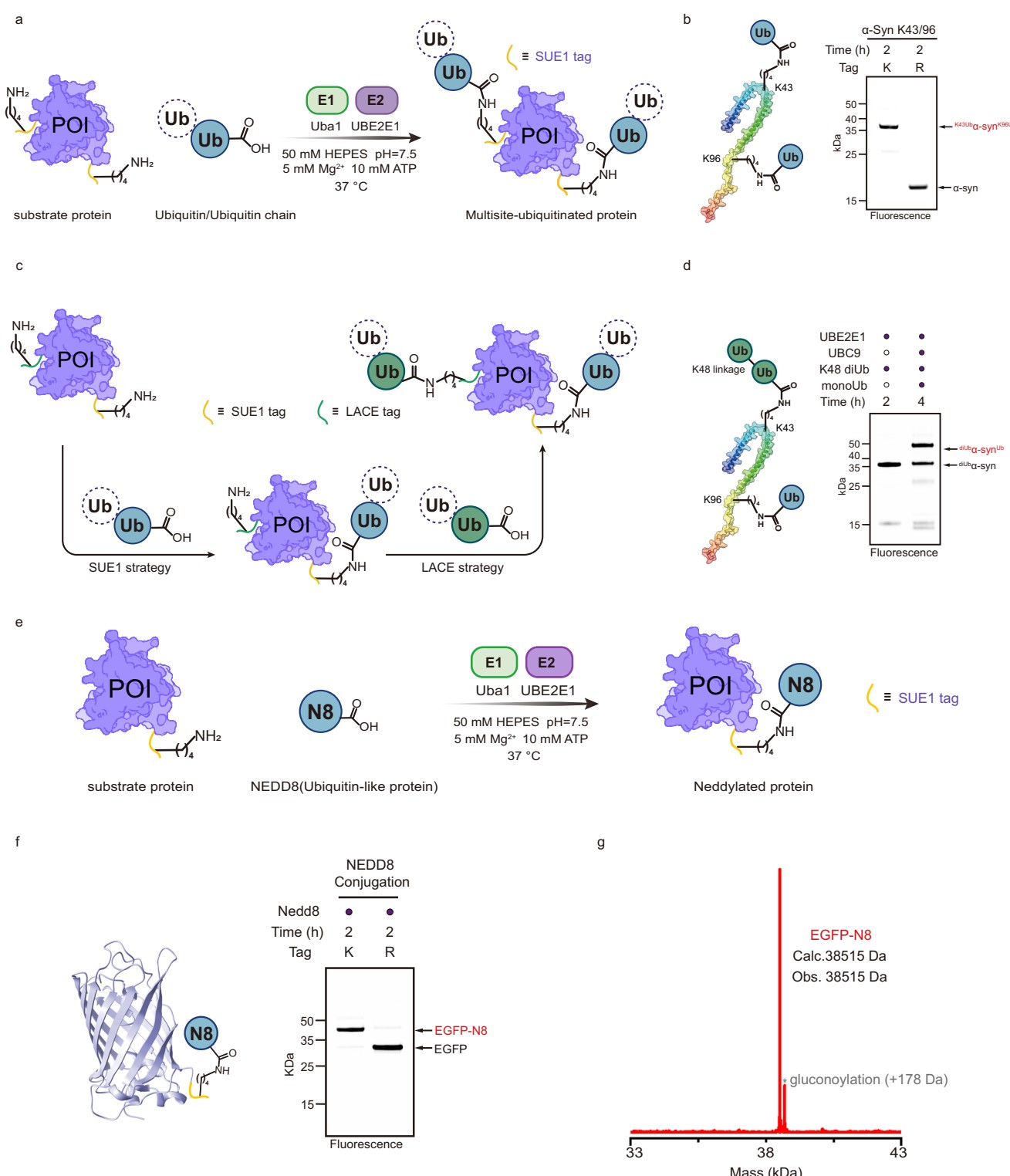

the protein. Then, we expressed and purified an α-synuclein mutant with a SUE1 tag introduced at Lys43 and a LACE tag introduced at Lys96. First, K48-linked diUb and the α-synuclein mutant with the same chemical equivalents were mixed, and then the diUb was transferred to Lys43 of α-synuclein with >90% conversion via our SUE1 strategy. Next, monoubiquitin, chimera E1 V4.5 (an engineered Ub-E1), and Ubc9 K14R[21] were added in situ to the reaction for monoubiquitination at Lys96. Finally, SDS–PAGE analysis showed the formation of dual-site differentially ubiquitinated α-synuclein with an overall yield of 59% (Fig. 5d).

In summary, SUE1 can be utilized alone or in combination with LACE to efficiently obtain multisite ubiquitinated proteins, providing an opportunity to investigate the role of multisite ubiquitination in regulating cellular events.

## SUE1-assisted synthesis of NEDD8-modified proteins

Inspired by the successful application of the SUE1 strategy in the preparation of various ubiquitinated substrates, we further examined whether our SUE1 strategy could generate ubiquitin-like modifications on protein substrates (Fig. 5e). We therefore performed the SUE1

**Fig. 5 | Application of SUE1 to defined multisite ubiquitinated proteins and site-specific NEDD8-modified proteins. a** Schematic representation of SUE1-mediated multisite ubiquitination, in which protein of interesting (POI) bears multiple (here two) SUE1 tags (KEGYEE) at customized sites. Generally, 1 μM Uba1, 20 μM UBE2E1, 8 μM substrate containing multiple SUE1 tags, 40 μM monoUb or Ub chain were mixed and reacted in the reaction buffer (50 mM HEPES, pH 7.5, 150 mM NaCl, 5 mM MgCl₂ and 10 mM ATP) at 37 °C for 2 h. **b** SDS–PAGE analysis of SUE1 mediated dual-site monoubiquitination at K43 and K96 of α-synuclein (α-Syn). **c** Schematic representation of the SUE1 strategy in combination with the LACE strategy for multi-site ubiquitinated proteins with different Ub chain linkages, in which protein of interesting (POI) bears the SUE1 tag and LACE tag at customized sites. **d** SDS–PAGE analysis of SUE1 and LACE co-mediated dual-site ubiquitination of α-synuclein (α-Syn) and K48-linked diUb modification at K43 by SUE1 strategy and monoubiquitination at K96 by LACE strategy. In general, 1 μM Uba1, 40 μM

UBE2E1, 16 μM substrate containing one SUE1 tag and one LACE tag and 16 μM Ub chain (or monoUb) were mixed and reacted in reaction buffer to achieve almost complete ubiquitination of substrate, and then 1 μM chimera E1 V4.5, 40 μM UBC9 K14R and 160 μM another Ub chain (or monoUb) were added to the reaction buffer at 37 °C for another 2 h. **e** Schematic representation of Uba1/UBE2E1-mediated site-specific neddylation, in which protein of interesting (POI) bears the SUE1 tag (KEGYEE) at the customized site. The reaction conditions were the same as for mono-ubiquitination, except NEDD8 (N8) was used to replace monoUb. **f** SDS–PAGE analysis of SUE1-Mediated NEDD8-modification of EGFP. **g** deconvoluted ESI-MS of the purified neddylated EGFP (EGFP-N8). Partially EGFP was gluconoylated on His tag (+178 Da) during expression in *E. coli*. Gel images shown in (**b**), (**d**) and (**f**) are representative of independent biological replicates (*n* = 2). Source data are provided as a Source Data file.

reaction on EGFP-C using several ubiquitin-like proteins, namely NEDD8, SUMO and ISG15, instead of Ub. Over 95% conversion of NEDD8 modification on lysine residues within the SUE1 tag was observed within 2 h, whereas there was no formation of SUMO-EGFP or ISG15-EGFP conjugation (Fig. 5f, g, Supplementary Fig. 8c). We also completed the NEDD8 modification of NCB1 at Lys64 via the SUE1 strategy with >90% conversion (Supplementary Fig. 8b). The compatibility of the SUE1 strategy with NEDD8 is consistent with previous report that NEDD8 can be activated by Uba1 with low kinetics, while Uba1 is incapable of activating SUMO/ISG15 and transferring it to UBE2E1[32].

### SUE1 facilitates biochemical evaluation of the K29/48-branched Ub chain to act as a signal for proteasomal degradation

The archetypal signal for proteasomal degradation consists of the attachment of a K48-linked Ub chain to a lysine residue in a target protein[33]. Interestingly, recent studies have shown that branched Ub chains, such as K29/48-branched Ub chains, elicit efficient protein degradation[34,35]. However, since branched ubiquitin chains are often produced cooperatively by multiple E3 enzymes, this makes the precise ubiquitin modification of substrate proteins more difficult to control[36]. In this context, whether the K29/48-branched Ub chain-modified substrate (here NCB1) could be degraded faster by the 26S proteasome than the K48-linked Ub chain-modified substrate of the same chain length remains unknown.

We used the SUE1 strategy to obtain the abovementioned ubiquitinated NCB1. Specifically, the SUE1 tag was introduced at K64 on NCB1 by mutations, and cysteine was introduced at the C-terminus for fluorescent labeling to track substrate degradation. For the preparation of K48-linked pentaUb-modified NCB1 (K48-Ub₅-NCB1), K48-linked pentaUb (preprepared by K48-specific E2 UBE2K) was directly transferred to NCB1 using the SUE1 strategy with 55% conversion and an isolated yield of 19% (Fig. 6a, Supplementary Fig. 9a, c). For K29/48-branched pentaUb-modified NCB1(K29/48-Ub₅-NCB1), we first transferred K29-linked triUb (preprepared by K29-specific E3 Ufd4) to NCB1 via the SUE1 strategy and then branched the K29-linked triUb chain at the K48 position using the E4 ligase Ufd2 as reported[34]. Finally, K29/48-Ub₅-NCB1 was obtained with an overall conversion of 38% and 11% isolated yield (Fig. 6a, Supplementary Fig. 9b, d).

With the above purified ubiquitinated NCB1 (Fig. 6b, Supplementary Fig. 10a), we performed proteasomal degradation experiments using the yeast 26S proteasome, whose composition and activity were verified by SDS–PAGE and in-gel activity assays (Supplementary Fig. 10b, c). As shown in Fig. 6c, the fluorescence intensity of K48-Ub₅-NCB1 decreased with increasing incubation time. The fluorescence intensity at 60 min was only 7% of the initial fluorescence intensity (Fig. 6d); while there was no significant change in fluorescence intensity in the absence of proteasomes (Supplementary Fig. 10d). In contrast, K29/48-branched pentaUb, although also targeting NCB1 to the proteasome for in

vitro degradation, has a reduced processing rate and requires twice the K48-linked pentaUb mediated degradation time to halve the fluorescence intensity of the substrate (Fig. 6c, d). These results suggest that the degradation of substrates triggered by the K29/48-branched Ub chain is less efficient than that triggered by the K48-linked Ub chain in vitro. Whether other cofactors/pathways exist in vivo to accelerate the efficiency of degradation triggered by the K29/48-branched Ub chain still needs further investigation.

## Discussion

Several previous structural studies have shaped our understanding of how E3 ligases (e.g., CUL1 complex, Ubr1) coordinate E2 enzymes for substrate recognition and ubiquitination[6,7]. However, it remains unclear how E2 enzymes directly mediate the site-specific ubiquitination of substrate proteins. In this study, we revealed an E3-independent mechanism of substrate ubiquitination in which the Ub E2 enzyme UBE2E1 can directly and specifically recognize and ubiquitinate a hexapeptide (KEGYES) derived from SETDB1. During typical ubiquitination, the core catalytic domain of the E2 enzyme, called the UBC domain, is responsible for binding to E1 and E3, enabling the E2 enzyme to act as an Ub-thioester carrier[18]. Here, we provide a unique case in which the UBC domain of UBE2E1 is capable of substrate recognition, a type of recognition that is sequence-dependent, and that recognition enables the transfer of Ub-thioester to the intrasequence lysine. Notably, mutations in the UBC domain of UbCH5c endow it with hexapeptide (KEGYES) recognition and the ability to ubiquitinate substrates independently of E3, broadening our understanding of the UBC domain of the E2 enzyme.

Obtaining precise ubiquitinated proteins is one of the keys to understanding the function of ubiquitination. Over the past 10 years, access to K48-linked ubiquitinated proteins has greatly improved our understanding of K48-linked ubiquitination-driven protein quality control through the proteasome and p97[37,38]. However, generating specific ubiquitinated substrates is by no means easy. Chemical approaches provide tools for the generation of some defined protein-Ub conjugates, but most of these chemical strategies are specialized techniques that are not readily accessible to a broad community of researchers, and some chemical operations and reagents cause protein unfolding[39–46]. In this context, chemoenzymatic approaches using enzymes (SUMO E2 Ubc9, Sortase, etc.) to site-specifically introduce ubiquitin or ubiquitin-like modifications on substrates have shown strong utility, without the aforementioned limitations[14,21,47,48]. In this work, guided by this catalytic structure of UBE2E1, we optimized a sequence (KEGYEE) with higher ubiquitination efficiency catalyzed by UBE2E1 and developed an E3-free strategy for the generation of ubiquitinated proteins with customized ubiquitinated sites, linkages, and lengths, termed SUE1. Moreover, our SUE1 strategy can work in combination with other chemical enzymatic methods, such as SUMO-

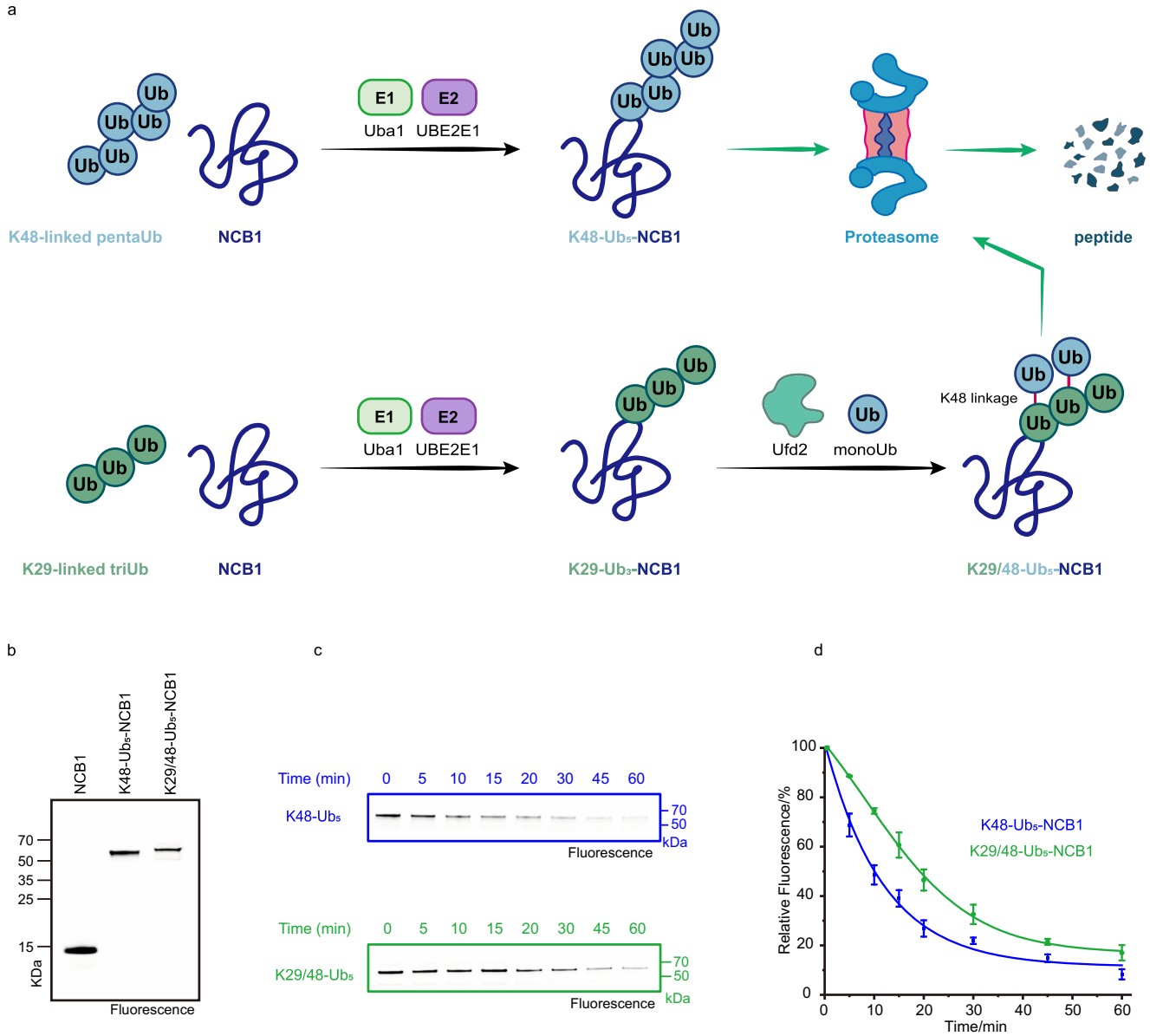

**Fig. 6 | SUE1 facilitates biochemical evaluation of proteasomal degradation signals. a** Schematic illustration of the route used to generate K48-linked or K29/48-branched pentaUb modified NCB1 for degradation assays. **b** In-gel-fluorescence analysis of ubiquitinated NCB1 and Coomassie-stained SDS–PAGE gel can be found in Supplementary Fig. 10. **c** Ubiquitinated NCB1 was mixed with purified 26S proteasomes for indicated time periods and degradation of substrates was detected by In-gel-fluorescence. **d** The average percentage of residual Ubiquitinated NCB1 at indicated time points after degradation (from **c**). Data represent the mean ± SD of three independent experiments. Gel images in (**b**) represent independent biological replicates ($n = 2$), and in (**c**) represent independent biological replicates ($n = 3$). Source data are provided as a Source Data file.

conjugating enzyme Ubc9-mediated ligation, which allows different forms of ubiquitination at different sites of the same protein. In addition, the SUE1 strategy is also applicable to ubiquitin-like NEDD8 modification, and to our knowledge, this is an unprecedented general enzymatic strategy for introducing NEDD8 modification into given proteins. However, in certain protein structural regions, recognition of SUE1 tag by UBE2E1 may be impeded, leading to decreased ubiquitination efficiency and, in some cases, discrepancies between the SUE1 tag and the native sequences of the substrate proteins may have a potential impact on the function of ubiquitinated proteins. In brief, we developed a sequence-dependent E3-free ubiquitination generation strategy capable of generating ubiquitinated substrates with native isopeptide bonds, highlighting the role of ubiquitin-related enzymes as an enzymatic tool library.

## Methods

### Cloning and plasmid construction

The human *UBE2E1* (NM_003341, https://www.ncbi.nlm.nih.gov/nuccore/NM_003341) cDNA was obtained from YouBio (Hunan, China) and the sequence corresponding to UBE2E1(41-193) carrying K136R to prevent autoubiquitination[49] was inserted into the vector pET28a-His containing an N-terminal His tag followed by an HRV 3C protease cleavage site. The gene encoding for enhanced green fluorescent protein (EGFP) with hexapeptide KEGYES fused to C-terminus was synthesized and subcloned into vector pET28a-His by GenScript (Nanjing, China). Variants of UBE2E1 and EGFP were generated using site-directed mutagenesis and all constructs were verified by DNA sequencing. The DNA sequence of Engineering human Ubch5c and *S. cerevisiae* Ubc4 were synthesized by GenScript and further cloned into the vector pET28a-His. The genes encoding for human α-synuclein,

human cyclin B1 (1-88, also known as NCB1) with His tag fused to C-terminus, and human NEDD8 were synthesized and subcloned into vector pET28a by GenScript. Variants of α-synuclein and NCB1 were generated using site-directed mutagenesis and all constructs were verified by DNA sequencing. The DNA sequence of human p53, *S. cerevisiae* Ufd2 and Ufd4 were synthesized and inserted between the BamHI and EcoRI sites of the vector pGEX-4T-1 with an N-terminal GST tag followed by an HRV 3C protease cleavage site by GenScript. The plasmids overexpressing human Uba1, human Ubiquitin, human M1-diUb, human UBE2K, human UBE2V1/UBE2N, chimera E1 V4.5 and human Ubc9 K14R were constructed as previously reported[6,20,21,50,51]. The amino acid sequences of proteins were listed in Supplementary Table 4.

### Protein expression and purification
Plasmids expressing the corresponding proteins were respectively transformed into *E. coli* BL21 (DE3) chemically competent cells and cultured in selective Luria Broth (LB) medium (Coolaber, Cat#PM0010-5kg) at 37 °C to $OD_{600}$ of 0.6. 0.5 mM isopropyl β-D-1-thiogalactopyranoside (IPTG, Purchased from LABLEAD Inc., Cat#0417) or 0.1 mM IPTG/50 μM $ZnCl_2$ in case of p53 was added to induced protein expression at 18 °C for 12–16 h. Cells were collected by centrifugation at $5000 \times g$ for 30 min at 4 °C and the precipitate was resuspended in 15 mL lysis buffer per liter cell culture (50 mM HEPES, pH 7.5, 150 mM NaCl). Cells were lysed by sonication and centrifuged at $17,418 \times g$ for 30 min at 4 °C, the supernatants or pellets containing proteins of interest were further purified. For proteins expressed using the vector pET28a-His, the supernatant was incubated with Ni-NTA affinity beads for 1–2 h at 4 °C, washed with lysis buffer, and then the proteins were eluted by lysis buffer containing 400 mM imidazole. For proteins expressed using the vector pGEX-4T-1, the supernatant was incubated with Glutathione affinity beads for 2 h at 4 °C, washed with lysis buffer, and then the proteins were eluted by lysis buffer containing 40 mM Glutathione. HRV 3C protease was added to the elution to remove the GST tag. Ion exchange and size-exclusion chromatography were used to further purify the proteins. α-synuclein, NCB1, Ub and NEDD8 were purified as previously described[14,52].

### In vitro E3-free Ubiquitination assay
E3-free ubiquitination assays were performed with 1 μM Uba1, 5 μM UBE2E1 or its variants or engineering Ubch5c, 80 μM Ub, and 8 μM peptide (KEGYES or variants)-fused EGFP or Ub as substrates in the reaction buffer (50 mM HEPES, pH 7.5, 150 mM NaCl, 5 mM $MgCl_2$ and 10 mM ATP) at 37 °C. These reactions were terminated by adding 4x sodium dodecyl sulfate (SDS) sample buffer containing 400 mM DL-Dithiothreitol (DTT), and then analyzed using SDS–PAGE and stained with Coomassie Brilliant blue.

The half-life of the Ubiquitination was estimated by a time-resolved ubiquitination kinetic assay, which was performed as described above, except that the reaction was sampled and quenched at the indicated time points (0, 0.5, 1, 1.5, 2, 3 h)[14]. Reaction conversion rate was estimated through densitometry analysis of Coomassie-stained bands by Image Lab-6.0.1 (Bio-Rad software) and conversion-time curves for ubiquitination were plotted.

### SUE1 strategy for generation of ubiquitinated (or neddylated) substrates
For the generating of site-specific ubiquitinated substrates, 1 μM Uba1, 20 μM UBE2E1, 8 μM substrate containing the SUE1 tag, 80 μM purified monoUb or NEDD8 or 40 μM Ub chain were mixed, reacted in the reaction buffer (50 mM HEPES, pH 7.5, 150 mM NaCl, 5 mM $MgCl_2$ and 10 mM ATP) at 37 °C, and the corresponding substrate containing the K to R mutation in the SUE1 tag was used in negative control experiments. For the transfer of K48-linked polyUb chains mixture prepared by UBE2K, 1 μM Uba1, 20 μM UBE2K, 80 μM monoUb were firstly mixed in the reaction buffer to product K48-linked polyUb chains mixture and then 20 μM UBE2E1 and 8 μM EGFP-C were added to the reaction buffer for transferring polyUb chains to EGFP-C.

These reactions were sampled by adding an equal volume of 4x sodium dodecyl sulfate (SDS) sample buffer containing 400 mM DL-dithiothreitol (DTT), then analyzed by SDS–PAGE, stained with Coomassie Brilliant Blue or imaged by fluorescence. For NEDD8-modified substrates, the reaction operated as described above, except that NEDD8 was used instead of ubiquitin or ubiquitin chains.

### Structure determination of UBE2E1 in complex with SETDB1-derived peptide
**Crosslinked complex generation and purification.** 130 μM active UBE2E1 mutant containing only one cysteine in the active center (UBE2E1 C Only) was incubated with 4-fold equivalents thiol-activated hexapeptide (AT-CEGYES) in the buffer (50 mM HEPES, pH 7.5, 150 mM NaCl) at 4 °C for 2 h and LC–MS confirmed that almost all UBE2E1 C Only formed disulfide-linked complexes with hexapeptide. The complex was further purified by size exclusion chromatography (Superdex 75 increase 10/300 GL column, GE Healthcare, Cat#29148721) pre-equilibrated with 25 mM HEPES, pH 7.5, 100 mM NaCl. Samples at indicated peaks were concentrated and LC–MS reconfirmed that the hexapeptide was linked to UBE2E1 C Only. It is worth noting that the complexes were prepared fresh and did not undergo freezing.

**Crystallization and structure determination.** UBE2E1/hexapeptide complex with a concentration of 3.5–4 mg mL$^{-1}$ was used for initial crystal screening based on the method of sitting-drop vapor diffusion at 18 °C. The complex was mixed with the crystallization reservoir at a volume ratio of 1:1 (0.5 μL: 0.5 μL) and Satisfying crystals grew under a reservoir solution containing 1.0 M Lithium sulfate, 0.1 M 2-Morpholinoethanesulphonic acid (MES), pH 6.5. Crystals were cryo-protected in the reservoir solution supplemented with a final concentration of 17% glycerol.

The diffraction of protein crystal was collected under the temperature of 100 K and wavelength of 0.979183 Å. The X-ray source was SSRF BEAMLINE BL02U1 of Shanghai Synchrotron Radiation Facility and the diffraction detector was DECTRIS EIGER2 S 9 M. Raw data was auto-processed and scaled with Aimless 0.7.7[53]. Molecular replacement phasing starting with the previous experimental model 5LBN[17] and structure refinement were carried out using PHENIX 1.19.2_4158[54]. The model building was visualized and operated by COOT-0.8.2[55]. Ramachandran statistics for our structure were 98.12 % favored, 1.88 % allowed, and 0.00 % outliers. Crystallographic data collection and refinement statistics are presented in Supplementary Table 1.

### Preparation of Ub chains
M1-diUb was obtained by *E. coli* expression by fusing two human ubiquitin genes. K6-diUb (Supplementary Fig. 13), K27-diUb (Supplementary Fig. 14), K11-diUb, K33-diUb and K11/48 branched triUb were obtained by chemical synthesis as we previously reported[22,23,56,57]. For the generation of K63-diUb, 1 μM Uba1, 20 μM UBE2V1/UBE2N and 1 mM Ub were mixed and reacted at 37 °C for 3–5 h in the reaction buffer (50 mM HEPES, pH 7.5, 150 mM NaCl, 5 mM $MgCl_2$ and 10 mM ATP). For the generation of K29-linked chains, 1 μM Uba1, 10 μM Ubc4, 2 μM Ufd4, and 1 mM Ub were mixed and reacted at 30 °C for 5–7 h in the reaction buffer. For the generation of K48-linked chains, 1 μM Uba1, 20 μM UBE2K and 1 mM Ub were mixed and reacted at 37 °C for 5–7 h in the reaction buffer. The enzymatic reactions were quenched by adding 5‰ Trifluoroacetic acid (TFA) and supernatants containing Ub chains were, respectively, separated by ion exchange (Mono S 5/50 cation exchange chromatography column, GE Healthcare, Cat#17516801) with buffer A containing 50 mM NaOAc, pH 4.5 and buffer B containing 50 mM NaOAc, pH 4.5, 1 M NaCl. Samples of the indicated peaks were collected and dialyzed into buffer (50 mM

HEPES, pH 7.5 and 150 mM NaCl), then concentrated and stored at −80 °C for further use.

## Assay to evaluate the orthogonality of SUE1 strategy and LACE strategy

To investigate the orthogonality of the SUE1 strategy to the substrate proteins of the LACE strategy, 1 μM Uba1, 20 μM UBE2E1, 8 μM EGFP containing LACE tag (LACE-tagged EGFP), 80 μM monoUb were mixed and reacted in reaction buffer (50 mM HEPES, pH 7.5, 150 mM NaCl, 5 mM MgCl$_2$ and 10 mM ATP) for 2 h. To verify the orthogonality of the LACE strategy to the substrate displaying the SUE1 tag, 1 μM chimera E1 V4.5, 20 μM Ubc9 K14R, 8 μM EGFP-C, and 80 μM monoUb were mixed and reacted for 2 h in reaction buffer. These reactions were sampled and an equal volume of 4x sodium dodecyl sulfate (SDS) sample buffer containing 400 mM DL-dithiothreitol (DTT) was added and then analyzed by SDS–PAGE, stained with Coomassie Brilliant Blue.

## Preparation of multisite ubiquitinated proteins

For multisite ubiquitinated α-synuclein with the same Ub chain linkages (here, monoUb), 1 μM Uba1, 20 μM UBE2E1, 8 μM Fluorescent-labeled α-synuclein containing two SUE1 tags, 40 μM monoUb were mixed and reacted in the reaction buffer (50 mM HEPES, pH 7.5, 150 mM NaCl, 5 mM MgCl$_2$ and 10 mM ATP) at 37 °C for 2 h. For multisite ubiquitinated α-synuclein with different Ub chain linkages (here K48-linked diUb modified at K43 site, monoUb modified at K96 site), 1 μM Uba1, 40 μM UBE2E1, 16 μM Fluorescent-labeled α-synuclein containing one SUE1 tag and one LACE tag and 16 μM K48- linked diUb were mixed and reacted in reaction buffer to achieve almost complete ubiquitination of α-synuclein (typically 2 h), and then 1 μM chimera E1 V4.5, 40 μM UBC9 K14R and 160 μM monoUb were added to the reaction buffer at 37 °C for another 2 h. These reactions were sampled and an equal volume of 4x sodium dodecyl sulfate (SDS) sample buffer containing 400 mM DL-dithiothreitol (DTT) was added and then analyzed by SDS–PAGE, imaged by fluorescence.

## Fluorescence labeling of substrates

α-synuclein, NCB1, and EGFP-C or their variants were fluorescently labeled to monitor the ubiquitination reaction or track substrate degradation. For the fluorescent labeling reaction, a 5-fold equivalents fluorescein-5-maleimide (Invitrogen, Cat#F150) were added to 50 μM purified protein and reacted at 30 °C for 2 h and then quenched by 2 mM DTT. Labeled proteins were further purified by size-exclusion chromatography (Superdex 200 increase 10/300 GL column, GE Healthcare, Cat#28990944) that pre-equilibrated with 50 mM HEPES, pH 7.5, 150 mM NaCl. Purified proteins were flash frozen in liquid nitrogen and stored at −80 °C for further use.

## Preparation of ubiquitinated NCB1 for proteasomal degradation assay

For the generation of K48-Ub$_5$-NCB1, 1 μM Uba1, 20 μM UBE2E1, 10 μM fluorescently labeled NCB1 containing the SUE1 tag, 12 μM purified K48 linked penta-Ub were mixed and reacted in the reaction buffer (50 mM HEPES, pH 7.5, 150 mM NaCl, 5 mM MgCl$_2$ and 10 mM ATP) at 37 °C for 3 h and final concentration 10 mM DTT was added. For the generation of K29/48-Ub$_5$-NCB1, 1 μM Uba1, 20 μM UBE2E1, 10 μM fluorescently labeled NCB1 containing the SUE1 tag, 12 μM purified K29 linked tri-Ub were mixed and reacted in the reaction buffer for 3 h and then 5 μM Ubc4 0.5 μM Ufd2 and 40 μM Ub were added to the reaction buffer to branch Ub chain at 30 °C for 30 min. The enzymatic reactions were respectively separated by size-exclusion chromatography (Superdex 200 increase 10/300 GL columnSuperdex 200 column, GE Healthcare, Cat#28990944) with buffer (50 mM HEPES, pH 7.5 and 150 mM NaCl), and target samples were collected and stored at −80 °C for further use.

## Purification of Yeast 26S Proteasome

The 26S proteasome was purified from the yeast strain (sDL66) with a cleavable protein A tag in subunit Rpn11 by an affinity purification method using rabbit IgG beads 4FF (Smart-Lifesciences, Cat# SA082005)[58]. The purity, integrity, and activity of the 26S proteasome were confirmed by SDS–PAGE, native PAGE and peptidase activity assay using Suc-LLVY-AMC according to standard protocols[28].

## In vitro protein degradation assay

30 nM yeast 26S proteasome and 800 nM ubiquitinated NCB1 were mixed in the proteasome assay buffer containing 10 mM Tris pH 7.4, 5 mM MgCl$_2$, 1% glycerol, 2 mM ATP and ATP-regeneration system (10 mM creatine phosphate and 2U creatine phosphokinase). The degradation assays were reacted at 30 °C and terminated at the indicated time points (0, 5, 10, 15, 20, 30, 45, 60 min) by 4x sodium dodecyl sulfate (SDS) sample buffer, then analyzed by SDS–PAGE imaged by fluorescence. Degradation effects were estimated through densitometry analysis of fluorescence bands by image lab (Bio-Rad software) and degradation efficiency curves were plotted.

## Reporting summary

Further information on research design is available in the Nature Portfolio Reporting Summary linked to this article.

## Data availability

The coordinates and structure factor files for UBE2E1-SETDB1 derived peptide complex have been deposited in the Protein Data Bank (PDB) under accession number 8IYA. The procedures and mass spectrometric characterization of synthesized proteins in this study is provided in the Supplementary Information. The HDX-MS and MS/MS generated in this study data have been deposited to the ProteomeXchange Consortium via the PRIDE partner repository with the dataset identifier PXD048115 and PXD048116. The atomic model of UBE2E1 is available under PDB accession code 3BZH and 5LBN. The atomic model of α-Synuclein is available under PDB accession code 1XQ8. The predicted atomic model of p53 is available in the AlphaFold Protein Structure Database under accession code AF-P04637-F1. Source data are provided with this paper.

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

## Acknowledgements

We thank the National Key R&D Program of China (No. 2022YFC3401500 for L. Liu, and 2023YFA0915300 for M. Pan) for financial support. This study was supported by the National Natural Science Foundation of China (22137005, 92253302, 22227810 for L. Liu, and 22277073 for M. Pan). L. Liu was supported by the XPLORER prize and the New Cornerstone Investigator Program. M. Pan was supported by the Shanghai Rising-Star Program (22QA1404900), Shanghai Pilot Program for Basic Research - Shanghai Jiao Tong University (21TQ1400224). We thank the X-ray crystallography platform of the Tsinghua University Technology Center for Protein Research and BL02U1 beam line of the Shanghai Synchrotron Research Facility for providing facility support.

## Author contributions

X. Wu, M. Pan and L. Liu proposed the idea and designed the experiments. X. Wu, L Liang, M. Pan and L. Liu analyzed all the results, wrote and revised the manuscript. X. Wu, R. Ding, T. Zhang and H. Cai carried out protein expression, purification. X. Wu, R Ding prepared ubiquitin chains. X. Wu, Y. Du, R. Ding prepared UBE2E1-hexapeptide complex and solved the crystal structure. X. Wu, L. Liang performed the ubiquitination and protein degradation assays. X. Tian and X. Wu performed the hydrogen-deuterium exchange experiment. All authors read and discussed the manuscript.

## Competing interests

The authors declare no competing interests.
