## [Peer Review File · Nature Communications]

REVIEWER COMMENTS

Reviewer #1 (Remarks to the Author):

In this manuscript, Wu et al. first focus on understanding the mechanism of E3-independent ubiquitination of SETDB1 by UBE2E1. The authors determined the crystal structure of UBE2E1 in complex with a chemically trapped substrate peptide and validated their structure via in vitro functional assays. The results guided design of a modified substrate peptide with higher E3-independent UBE2E1-mediated ubiquitination efficiency and the authors demonstrate that introduction of this modified peptide sequence into test proteins resulted in ubiquitination at the desired site with customized ubiquitin chain linkages and lengths (with the caveat of needing to mutate the substrate protein sequence around the target lysine to accommodate the SUE1 approach). Wu et. al further demonstrate that this strategy can also be used for multisite ubiquitination either by itself or in combination with the previously published LACE approach, with the latter approach allowing for generation of different forms of ubiquitination at different sites of the same protein. Overall, the authors have successfully developed an enzymatic method for customized ubiquitination (or NEDDylation) of proteins at desired sites with native isopeptide bonds which should be a useful tool for studying the effects of target protein ubiquitination in vitro. With that said, there are a few important issues that should be addressed to make this manuscript an even stronger candidate for publication in Nature Communications:

- 1) The authors have developed a useful tool for customized ubiquitination at desired sites in vitro. It would be very interesting to see whether the authors can extend the application of this tool to a cellular setting which could further increase the impact of this work. In this regard, are the KEGYES or KEGYEE sequences present in any other human proteins?
- 2) The authors should expand the sequence alignment presented in Supplementary Figure 2F and provide some more discussion on the specificity of UBE2E1 for this target sequence
- 3) Regarding mechanism, presumably Asp163 UBE2E1 is involved in deprotonation of the incoming lysine nucleophile? The authors should mutate this residue and test it in their in vitro ubiquitination reaction to validate the mechanism.
- 4) The rationale for studying this system in paragraph 3 of the introduction is somewhat weakly laid out, particularly in the context of the focus of the work throughout the majority of the manuscript
- 5) The authors should carefully proofread their manuscript as there are several typos throughout (e.g. line 30- branchedly ubiquitylated).

Reviewer #2 (Remarks to the Author):

Summary:

This manuscript describes the mechanism of an E3-independent ubiquitin transfer by elucidating the structure of a human ubiquitin E2 enzyme UBE2E1 in complex with a short peptide derived from its substrate. Optimization of the peptide sequence guided by structural insights enabled the use of the peptide as a general ubiquitination tag KEGYEE. The authors further showcase the E3-independent ubiquitination strategy, SUE1, by demonstrating the transfer of ubiquitin monomers and oligomers including branched chains and NEDD8. This strategy offers orthogonality to the previously reported LACE strategy (Hofmann et al., Nat. Chem. 2020, 1008; Akimoto et al., ACS Cent. Sci. 2022, 275), thereby expanding the tool box of enzymatic site-specific ubiquitination. I recommend the publication of this manuscript after minor improvements.

Comment #1 – Are there structural limitations for the placement of the SUE1 tag? In figure 4b, the lysine residue appears to be in an α -helix. I assume the crystal structure represents the wild type protein and wonder if the secondary structure is still retained after the mutations needed to create a SUE1 tag.

Comment #2 – Other points mainly regarding formatting can be found below.

Line 30: Throughout the manuscripts, the terms “ubiquitination”, “ubiquitinated”, “ubiquitylation” and “ubiquitylated” are used inconsistently.

Line 70: “NEDD8” and “Nedd8” are used inconsistently.

Line 244: Wrong names for chimeric E1 v4.5 and Ubc9 K14R. Also, chimeric E1 v4.5 is rather an engineered Ub E1 than an engineered SUMO E1.

Line 246: It might be worth noting that the two methods are not completely orthogonal. If Ub remained in the first step, it can be transferred in the second step, and if a portion of SUE1 tag remain unreacted in the first step, the remaining SUE1 enzymes can still transfer the Ub for the second step. It is impressive

the authors achieved these two reactions without a purification step in between. It can require some optimizations in reaction conditions.

Line 327: “even for K63-linked ubiquitinated substrates.” I personally find this difficult to interpret. There may be some logical steps missing. K63 chains are known to be easier to transfer site-specifically?

Figures in general: For the presented reactions, it would be helpful for the readers if reaction conditions were presented either in the main text or in the figure captions. Additionally, some fluorescence SDS-PAGE images appear to be over-processed and have unnecessarily high contrast, making some minor bands difficult to see.

Figure 2c: The structures and interactions presented in the figure seem to be wrong. L165 is shown as an ester, S168 should interact with hexapeptide S6, and S126 should interact with hexapeptide S5. Similar mistakes are seen in extended figure 2d as well.

Figure 3: In the figure and the main text, terms diUb, 2Ub, Ub2 are used seemingly inconsistently.

Extended figure 7a: Labellings on top of the image are not aligned with the columns.

Reviewer #3 (Remarks to the Author):

In this manuscript Liu and coworkers reported an exciting new strategy in achieving E3-independent protein ubiquitination by utilizing the E2 UBE2E1. This work was based on a structural and biochemical characterization of UBE2E1 bound with a hexapeptide derived from SETDB1, a known substrate of the E2. This cocrystal structure enables the subsequent mutational analysis of the hexapeptide that eventually led to the improved SUE1 tag for sequence-dependent and E3-independent enzymatic ubiquitination. Using the SUE1 tag, the authors demonstrated that this method can be applied to a number of different proteins and a range of ubiquitin chain structures. This is an impressive amount of work showcasing the versatility of the method. The manuscript is well-written and should be of interest to readers of Nature Communications, while also offering a useful tool for researchers in the ubiquitin field. Below are some points for the authors to consider in order to further enhance an already well-crafted manuscript.

1. The S6E mutation of UBE2E1 was found to increase the catalytic efficiency of ubiquitination significantly. The kinetic data presented in SI Fig. 3C seems to suggest that a decrease in K_m contributes to the higher catalytic efficiency. The authors should look into this by fitting their time-dependent ubiquitination data to obtain K_m values. Related to this is the question how tight the bindings of the original hexapeptide and the S6E peptide to UBE2E1 are. Have the authors measured this? A sufficiently high binding affinity of the peptide to UBE2E1 may be a key feature of E3-independent ubiquitination by E2.

2. One important application of the UBE2E1-catalyzed ubiquitination is to generate homogeneous ubiquitinated POI with sufficient quantity and purity as exemplified by the preparation of ubiquitinated NCB1. It appears that in the enzymatic reactions a molar excess of UBE2E1 was used relative to the substrate protein. Given UBE2E1 is catalytic, why is this necessary? Lowering the amount of UBE2E1 used should make this method more accessible and efficient.

3. Throughout the manuscript, the authors reported the percentage of conversion in all ubiquitination reactions. Considering that purification is often required to obtain the final ubiquitinated protein, including a final yield of the ubiquitinated protein product such as ubiquitinated NCB1 will be informative to the readers.

4. The authors should comment on the deviation of the SUE1 tag sequence from the native sequence at the site of ubiquitination and its potential impact on the function of the ubiquitinated proteins. Could this be a potential contribution to the lack of stimulation of NCB1 degradation by the K29/K48 branch compared to the K48 chain.

Reviewer #4 (Remarks to the Author):

Following are the detailed responses to reviewers' comments:

Reviewer #1 :

In this manuscript, Wu et al. first focus on understanding the mechanism of E3-independent ubiquitination of SETDB1 by UBE2E1. The authors determined the crystal structure of UBE2E1 in complex with a chemically trapped substrate peptide and validated their structure via in vitro functional assays. The results guided design of a modified substrate peptide with higher E3-independent UBE2E1-mediated ubiquitination efficiency and the authors demonstrate that introduction of this modified peptide sequence into test proteins resulted in ubiquitination at the desired site with customized ubiquitin chain linkages and lengths (with the caveat of needing to mutate the substrate protein sequence around the target lysine to accommodate the SUE1 approach). Wu et. al further demonstrate that this strategy can also be used for multisite ubiquitination either by itself or in combination with the previously published LACE approach, with the latter approach allowing for generation of different forms of ubiquitination at different sites of the same protein. Overall, the authors have successfully developed an enzymatic method for customized ubiquitination (or NEDDylation) of proteins at desired sites with native isopeptide bonds which should be a useful tool for studying the effects of target protein ubiquitination in vitro.

Our response: We thank the reviewer for the objective summary and positive comments.

With that said, there are a few important issues that should be addressed to make this manuscript an even stronger candidate for publication in Nature Communications:

1) The authors have developed a useful tool for customized ubiquitination at desired sites in vitro. It would be very interesting to see whether the authors can extend the application of this tool to a cellular setting which could further increase the impact of this work. In this regard, are the KEGYES or KEGYEE sequences present in any other human proteins?

Our response: We appreciate the reviewer's suggestion, and have utilized the BLAST service website (<https://blast.ncbi.nlm.nih.gov/>) to retrieve KEGYES/KEGYEE sequences in the Homo

sapiens (taxid:9606) standard database (Non-redundant protein sequences), based on the website's default adjustment of parameters for short input sequences. In addition to SETDB1 and its isoforms, the original sequence KEGYES was only found in the immunoglobulin heavy chain junction region (**Rev. Table. 1**), which appears as a non-cytoplasmic protein, whereas the optimal KEGYEE sequence was not detected in any human proteins.

Rev. Table. 1 KEGYES sequence in human proteins.

Description	Total Score	Query Cover	E value	Per. ident	Acc. Len	Accession
immunoglobulin heavy chain junction region	22.3	100%	12	100%	12	MBB2055 263.1
histone-lysine N-methyltransferase SETDB1 isoform 4	22.3	100%	18	100%	1292	NP_00135 3346.1
histone-lysine N-methyltransferase SETDB1 isoform 1	22.3	100%	18	100%	1291	NP_00113 8887.1
histone-lysine N-methyltransferase SETDB1 isoform 5	22.3	100%	18	100%	1291	NP_00138 0890.1
SET domain, bifurcated 1	22.3	100%	18	100%	1290	AAH2867 1.1
histone-lysine N-methyltransferase SETDB1 isoform 2	22.3	100%	18	100%	1290	NP_03656 4.3
SET domain, bifurcated 1, isoform CRA_b	22.3	100%	18	100%	1173	EAW5350 4.1
histone-lysine N-	22.3	100%	18	100%	738	XP_04729

#Data was available by BLAST service website. The description column and the accession column are the name and genebank number, respectively, of the protein that includes the target sequence, and the other columns are score messages.

Considering the uniqueness of the KEGYEE sequence and the fact that the enzyme UBE2E1 and Uba1 required for the SUE1 reaction is naturally present in human cells, the SUE1 reaction may be a useful tool for studying ubiquitination in vivo. However, because deubiquitinase-mediated reverse deubiquitination is widespread, the development of this application requires further investigation. Related research is underway in our and collaborators' laboratories and will be shared with the research community in due course.

2) The authors should expand the sequence alignment presented in Supplementary Figure 2F and provide some more discussion on the specificity of UBE2E1 for this target sequence.

Our response: We thank the reviewer's comment. We have expanded the comparison of the sequence of UBE2E1 with UbCH5c, UBCH7, UBE2S, UBE2K and UBE2N, UBE2W, UBE2R1 and Ube2T, etc., and this result is updated in Extended Figure 2f. Based on the unique amino residues (N125/S126/P164/L165/G167/S168) within UBE2E1, we discussed in the main text why UBE2E1 specifically recognizes and ubiquitinates proteins containing target sequences (e.g., SETDB1), whereas other E2 enzymes do not. In addition, targeted mutation of amino residues in Ubch5c to enable SUE1 reaction further validates our conclusion (Figure. 2e).

Extended Figure 2f. The complete interaction network required for the recognition of hexapeptides is specific to UBE2E1. E2s, with different preferences for ubiquitin chain linkages, were selected to compare with UBE2E1.

3) Regarding mechanism, presumably Asp163 UBE2E1 is involved in deprotonation of the incoming lysine nucleophile? The authors should mutate this residue and test it in their *in vitro* ubiquitination reaction to validate the mechanism.

Our response: We thank the reviewer's suggestion. We tested the UBE2E1 D163A mutant for ubiquitination activity (**Rev. Figure. 1**), and indeed the mutation of D163A dramatically diminished UBE2E1's ability to ubiquitinate substrate, as can be found in Extended Figure 2b.

Rev.Figure.1 *In vitro* E3-free Ubiquitination assay using UBE2E1 mutants with EGFP* (a model substrate bearing the hexapeptide KEGYES) as substrates. UBE2E1 D163A mutant's result is highlighted using a red box.

4) The rationale for studying this system in paragraph 3 of the introduction is somewhat weakly laid out, particularly in the context of the focus of the work throughout the majority of the manuscript

Our response: We thank the reviewers for their suggestions and have revised the logic of paragraph 3 as follows.

“Given that identifying the E3s for target proteins is a key challenge in reconstituting ubiquitinated substrates, as described above, elucidating the underlying mechanism of this extraordinary E3-free Ub cascade occurring in human cells not only will improve our understanding of the ubiquitination reaction evolved from nature, but also may provide useful strategies for the acquisition of ubiquitinated proteins¹². To this end, we focused on a unique human ubiquitin-conjugating enzyme E2 E1 (UBE2E1) that can catalyze monoubiquitination at K867 of its substrate protein SETDB1 independently of E3 through an unknown mechanism¹³.”

5) The authors should carefully proofread their manuscript as there are several typos throughout (e.g. line 30- branchedly ubiquitylated).

Our response: We thank the reviewer for pointing out the typos. We have carefully/thoroughly fixed these errors.

Reviewer #2:

Summary: This manuscript describes the mechanism of an E3-independent ubiquitin transfer by elucidating the structure of a human ubiquitin E2 enzyme UBE2E1 in complex with a short peptide derived from its substrate. Optimization of the peptide sequence guided by structural insights enabled the use of the peptide as a general ubiquitination tag KEGYEE. The authors further showcase the E3-independent ubiquitination strategy, SUE1, by demonstrating the transfer of ubiquitin monomers and oligomers including branched chains and NEDD8. This strategy offers orthogonality to the previously reported LACE strategy (Hofmann et al., Nat. Chem. 2020, 1008; Akimoto et al., ACS Cent. Sci. 2022, 275), thereby expanding the tool box of enzymatic site-specific ubiquitination. I recommend the publication of this manuscript after minor improvements.

Our response: We thank the reviewer's positive and constructive comments on our work.

Comment #1 – Are there structural limitations for the placement of the SUE1 tag? In figure 4b, the lysine residue appears to be in an α -helix. I assume the crystal structure represents the wild type protein and wonder if the secondary structure is still retained after the mutations needed to create a SUE1 tag.

Our response: The SUE1 reaction requires UBE2E1 to recognize the SUE1 tag, and in certain structural regions of some proteins, this recognition may be impeded, which may lead to less efficient ubiquitination of these substrates. To make this clearer, we added a note (“...*in certain protein structural regions, recognition of SUE1 tag by UBE2E1 may be impeded, leading to decreased ubiquitination efficiency*”) in the discussion section. Exploring the secondary structure of SUE1 tag-containing substrates usually requires high-resolution structures, which is often time-consuming and not easy. We do not have any experimental evidence to prove that the secondary structure of p53 containing the SUE1 tag at Lys24 (in Figure. 4b) is or is not altered compared to wild-type p53. To avoid misunderstanding by the reader, we have stated in the figure captions (in Figure. 4) that the structural model of the substrate is the wild-type protein, derived from Protein Data Bank or AlphaFold Protein Structure Database.

Comment #2 – Other points mainly regarding formatting can be found below.

Line 30: Throughout the manuscripts, the terms “ubiquitination”, “ubiquitinated”, “ubiquitylation” and “ubiquitylated” are used inconsistently.

Line 70: “NEDD8” and “Nedd8” are used inconsistently.

Line 244: Wrong names for chimeric E1 v4.5 and Ubc9 K14R. Also, chimeric E1 v4.5 is rather an engineered Ub E1 than an engineered SUMO E1.

Line 327: “even for K63-linked ubiquitinated substrates.” I personally find this difficult to interpret. There may be some logical steps missing. K63 chains are known to be easier to transfer site-specifically?

Our response: We thank the reviewer for pointing out these out, and corrected them as requested. In addition, for clarity, we deleted "even for K63-linked ubiquitinated substrates." in Line327.

Line 246: It might be worth noting that the two methods are not completely orthogonal. If Ub remained in the first step, it can be transferred in the second step, and if a portion of SUE1 tag remain unreacted in the first step, the remaining SUE1 enzymes can still transfer the Ub for the second step. It is impressive the authors achieved these two reactions without a purification step in between. It can require some optimizations in reaction conditions.

Our response: We agree with the reviewer and we indeed optimized the reaction conditions, which can be found in the Methods. Briefly, in the first step of the ubiquitination reaction, K48-linked diUb of equal stoichiometric ratio to the substrate α -synuclein mutant was used to avoid excess K48-linked diUb remaining after the ubiquitination modification. In the second step of the reaction, monoUb with a stoichiometric ratio of 10-fold with respect to the substrate was used, to facilitate the mono-ubiquitin modification for the second ubiquitination step.

Figures in general: For the presented reactions, it would be helpful for the readers if reaction conditions were presented either in the main text or in the figure captions.

Additionally, some fluorescence SDS-PAGE images appear to be over-processed and have unnecessarily high contrast, making some minor bands difficult to see.

Our response: We thank the reviewer for suggestions regarding the Figure. We added the reaction conditions in the figure captions (Figure. 3a, Figure. 5a/c/e). We also adjusted the contrast of the fluorescence SDS-PAGE images (Figure. 5d, f) according to Nature Communications' image integrity and standards.

Figure 2c: The structures and interactions presented in the figure seem to be wrong. L165 is shown as an ester, S168 should interact with hexapeptide S6, and S126 should interact with hexapeptide S5. Similar mistakes are seen in extended figure 2d as well.

Figure 3: In the figure and the main text, terms diUb, 2Ub, Ub2 are used seemingly inconsistently.

Extended figure 7a: Labellings on top of the image are not aligned with the columns.

Our response: We thank the reviewer for pointing out these errors, and have corrected them as suggested. Furthermore, we also rectified the numbering of amino acid sequences in Figure 2e.

Reviewer #3:

In this manuscript Liu and coworkers reported an exciting new strategy in achieving E3-independent protein ubiquitination by utilizing the E2 UBE2E1. This work was based on a structural and biochemical characterization of UBE2E1 bound with a hexapeptide derived from SETDB1, a known substrate of the E2. This cocrystal structure enables the subsequent mutational analysis of the hexapeptide that eventually led to the improved SUE1 tag for sequence-dependent and E3-independent enzymatic ubiquitination. Using the SUE1 tag, the authors demonstrated that this method can be applied to a number of different proteins and a range of ubiquitin chain structures. This is an impressive amount of work showcasing the versatility of the method. The manuscript is well-written and should be of interest to readers of Nature Communications, while also offering a useful tool for researchers in the ubiquitin field. Below are some points for the authors to consider in order to further enhance an already well-crafted manuscript.

Our response: We thank the reviewer for the positive comments.

1. The S6E mutation of UBE2E1 was found to increase the catalytic efficiency of ubiquitination significantly. The kinetic data presented in SI Fig. 3C seems to suggest that a decrease in K_m contributes to the higher catalytic efficiency. The authors should look into this by fitting their time-dependent ubiquitination data to obtain K_m values. Related to this is the question how tight the bindings of the original hexapeptide and the S6E peptide to UBE2E1 are. Have the authors measured this? A sufficiently high binding affinity of the peptide to UBE2E1 may be a key feature of E3-independent ubiquitination by E2.

Our response: We thank the reviewer's suggestion, and have measured the K_m , K_{cat} values of UBE2E1 towards the original hexapeptide and the S6E peptide using a standard protocol based on Lineweaver-Burk plots (**Rev. Figure. 2a**). The results showed that UBE2E1 exhibits similar K_m for both peptide substrates (slightly lower for the original hexapeptide), whereas the K_{cat} for the S6E peptide was 2.5-fold higher than that for the original hexapeptide. (**Rev. Table. 2**).

We attempted to measure the affinity of the two peptides for UBE2E1 separately by isothermal titration calorimetry (ITC) analysis using a MicroCal ITC 200 instrument, but

unfortunately no significant binding signal was detected under the following conditions: 30 μM UBE2E1 in the sample cell was titrated with 500 μM KEGYEE or KEGYES peptide through 19 injections (2.0 μl each) at 25 $^{\circ}\text{C}$ and 750 r.p.m. stirring speed. We propose this phenomenon is mainly due to the weak affinity between the peptide and UBE2E1. Similarly, the affinity of UBC9 for SUMO motif-containing peptides (p53 or c-Jun peptide) is too weak to measure, estimated to be in 3-6 mM [Journal of Biological Chemistry 277.24 (2002): 21740-21748].

To further elucidate the interaction between the peptide and UBE2E1, we performed hydrogen-deuterium exchange mass spectrometry (HDX-MS). The UBE2E1 was subjected to a 90-second deuterium exchange reaction in the presence or absence of the KEGYES peptide, followed by pepsin digestion and mass spectrometry analysis. By analyzing and comparing the deuterium exchange degrees of UBE2E1 in both cases with and without the peptide, the region of interaction between the UBE2E1 and the peptide was revealed. We observed a significant decrease in the deuterium exchange level of the UBE2E1(116-132) fragment relative to the absence of the peptide, indicating that this region interacts with the peptide (**Rev. Figure. 2b/c**). Notably, UBE2E1 (116-132) fragment is close to the catalytic cysteine and contains the key amino acid $^{125}\text{N}^{126}$ for the interaction observed in the crystal structure of the complex. Thus, HDX-MS, structural and biochemical results indicate the peptide interacts with UBE2E1, although the affinity is weak, sufficient for the ubiquitination in the absence of the E3 enzyme.

Rev. Table. 2 Kinetic parameters of UBE2E1-mediated Ubiquitination for original hexapeptide and the S6E peptide.

Substrate	K_{cat} (min^{-1})	K_{m} (μM)	$K_{\text{cat}}/K_{\text{m}}$ ($\text{M}^{-1}\text{s}^{-1}$)
KEGYEE	1.13	491.88	38.29
KEGYES	0.45	382.70	19.60

Rev. Figure. 2. Enzymatic and Interaction Characterization of UBE2E1

a, Double reciprocal Lineweaver–Burk’s plots of UBE2E1 using KEGYEE or KEGYES sequence as a substrate. **b**, HDX-MS difference map of UBE2E1 in the presence and absence of the peptide (KEGYES) at deuterium exchange time of 90 seconds. Changes in deuterium incorporation corresponding to the UBE2E1 (116-132) fragment are highlighted by red shade. **c**, HDX-MS results of UBE2E1 (116-132) in the presence and absence of the peptide (KEGYES). The centroid mass changes between the two states as indicated in the red arrow.

2. One important application of the UBE2E1-catalyzed ubiquitination is to generate homogeneous ubiquitinated POI with sufficient quantity and purity as exemplified by the preparation of ubiquitinated NCB1. It appears that in the enzymatic reactions a molar excess of UBE2E1 was used relative to the substrate protein. Given UBE2E1 is catalytic, why is this necessary? Lowering the amount of UBE2E1 used should make this method more accessible and efficient.

Our response: We thank the reviewer’s comments. Although lower concentrations (5 μM) of UBE2E1 is still able to catalyze the ubiquitination reaction, we found that the use of excess UBE2E1 (20 μM) can shorten the ubiquitination reaction time (**Rev. Figure. 3**), which was also demonstrated in the “LACE” approach. Meanwhile, UBE2E1 can be obtained in large quantities (>10 mg/L) through the *E. coli* expression system, and in principle, can be easily removed by Ni-NTA after ubiquitination reaction.

Rev. Figure. 3 The ubiquitination conversion over time at 5 or 20 μ M UBE2E1.

3. Throughout the manuscript, the authors reported the percentage of conversion in all ubiquitination reactions. Considering that purification is often required to obtain the final ubiquitinated protein, including a final yield of the ubiquitinated protein product such as ubiquitinated NCB1 will be informative to the readers.

Our response: We thank the reviewer's suggestion, and have added the final isolated yield data for ubiquitinated NCB1 in the main text ("*...with 55% conversion and an isolated yield of 19%.*" and "*...an overall conversion of 38% and 11% isolated yield.*")

4. The authors should comment on the deviation of the SUE1 tag sequence from the native sequence at the site of ubiquitination and its potential impact on the function of the ubiquitinated proteins. Could this be a potential contribution to the lack of stimulation of NCB1 degradation by the K29/K48 branch compared to the K48 chain.

Our response: We thank the reviewer's suggestion, and have added comments as suggested to the discussion section ("*...in some cases, discrepancies between the SUE1 tag and the native sequences of the substrate proteins may have a potential impact on the function of ubiquitinated proteins*"). Given that the K48 chain and the K29/K48 branch chain were modified on NCB1 with the same sequence (containing SUE1 tag), we propose that the difference in degradation efficiency between them is mainly due to the difference in modified ubiquitin chain type, and is less likely to be caused by the deviation of the SUE1 tag sequence from the native sequence.

Reviewer #4:

Our response: We thank the reviewer's contribution to the peer review.

REVIEWERS' COMMENTS

Reviewer #3 (Remarks to the Author):

The authors have satisfactorily addressed all the points raised in my initial review. The only suggestion for the authors is to consider incorporating Review Table 2 and Figures 2 into the final version of the manuscript, as they provide valuable information and further enhance the paper.

Reviewer #3 (Remarks to the Author):

The authors have satisfactorily addressed all the points raised in my initial review. The only suggestion for the authors is to consider incorporating Review Table 2 and Figures 2 into the final version of the manuscript, as they provide valuable information and further enhance the paper.

Our response: We thank the reviewer's suggestion. We have added Rev. Table. 2 and Rev. Figure. 2 to the final manuscript and renamed them Supplementary Table 2 and Supplementary Figure 3. The relevant description has also been added to the main text:

The K_m , k_{cat} values of UBE2E1 towards the original hexapeptide and the S6E peptide were further measured using a standard protocol based on Lineweaver-Burk plots (Supplementary Table 2), and the kinetic data showed that the S6E mutation increased the ubiquitination activity from 19.60 to 38.29 $M^{-1} s^{-1}$ and shortened the half-life of the reaction by 2.9-fold (from 1.36 hours to 0.47 hours) (Supplementary Figure 4c).

Additionally, Hydrogen-deuterium exchange mass spectrometry (HDX-MS) was employed to further validate the interaction between the UBE2E1 and the hexapeptide in solution. In the presence of the hexapeptide, a significant decrease in the deuterium exchange level was observed for the UBE2E1 (116-132) fragment (Supplementary Fig.3), which is close to the catalytic cysteine and contains the key amino acids N125/S126 for the interaction observed in the crystal structure.